# Automatic Defects Recognition of Lap Joint of Unequal Thickness Based on X-Ray Image Processing

**DOI:** 10.3390/ma17225463

**Published:** 2024-11-08

**Authors:** Dazhao Chi, Ziming Wang, Haichun Liu

**Affiliations:** 1National Key Laboratory of Precision Welding and Joining of Materials and Structures, Harbin Institute of Technology, Harbin 150001, China; 22s009128@stu.hit.edu.cn; 2PipeChina Engineering Quality Supervision and Inspection Company, Beijing 100013, China; liuhc@pipechina.com.cn

**Keywords:** lap joint, non-destructive testing, X-ray, image processing, defect detection

## Abstract

It is difficult to automatically recognize defects using digital image processing methods in X-ray radiographs of lap joints made from plates of unequal thickness. The continuous change in the wall thickness of the lap joint workpiece causes very different gray levels in an X-ray background image. Furthermore, due to the shape and fixturing of the workpiece, the distribution of the weld seam in the radiograph is not vertical which results in an angle between the weld seam and the vertical direction. This makes automatic defect detection and localization difficult. In this paper, a method of X-ray image correction based on invariant moments is presented to solve the problem. In addition, a novel background removal method based on image processing is introduced to reduce the difficulty of defect recognition caused by variations in grayscale. At the same time, an automatic defect detection method combining image noise suppression, image segmentation, and mathematical morphology is adopted. The results show that the proposed method can effectively recognize the gas pores in an automatic welded lap joint of unequal thickness, making it suitable for automatic detection.

## 1. Introduction

Welding as a manufacturing technique has been widely used in industry. In order to avoid structural failure, it is necessary to conduct a reliability inspection of welding quality [1]. X-ray examination has become prevalent across different types of welding (arc welding, fusion welding, etc.), additive manufacturing, and numerous other manufacturing processes to fulfill defect detection and evaluation [2,3,4]. As these industries undergo rapid advancements [5], X-ray inspection proves invaluable due to its capacity to yield immediate and intuitive outcomes, facilitating the swift identification of volumetric flaws [6,7,8]. The manual method of reviewing X-ray films is a common way to conduct quality examinations for welding. Images consist of the inside of the inspected specimen, which is obtained from inspection devices [9]. At present, the commonly used X-ray testing technologies include the film method, real-time imaging method, and X-ray computed tomography (CT) [10,11,12]. Techniques based on X-ray inspection are the most intuitive approach for non-destructive testing (NDT), material evaluation and residual stresses analysis of flat weldment [13,14,15,16], and pipeline [17,18], helping to avoid safety accidents and evaluate the quality of welds.

In the area of X-ray inspection evaluation, manual evaluation is currently the primary method employed, which imposes high technical requirements on the inspectors. Furthermore, prolonged work can lead to visual fatigue among inspectors, making this method unsuitable for evaluating large volumes of radiographic films. To address the need for efficient evaluation, automatic defect recognition technology has emerged. In recent years, defect detection based on machine recognition has seen enormous progress. For example, the application of Support Vector Machines (SVM) in weld defect detection of X-ray images is receiving more and more attention, which has led to the rapid development of weld automatic detection [19,20,21]. At the same time, the idea of using neural networks has been widely accepted, and a large amount of research has been conducted based on deep learning in medicine, particularly in human brain function [22,23]. As a result, automatic defect recognition methods based on neural networks, such as Convolutional Neural Networks (CNNs) [24,25] and the dual-graph interactive consistency reasoning network (DGICR-Net) [26], have been researched. These methods are conducive to improving the reliability of weld defect detection in X-ray images. However, due to the complexity and limitations of imaging conditions, objects, and algorithms, there are still many problems to be further studied regarding practical applications [27,28]. In particular, when the welded structure to be detected is complex, higher requirements are necessary for the detection.

Automatic recognition of defects in radiographic inspection of lap welds on plates of unequal thickness remains challenging. This is primarily due to variations in the thickness of the workpiece, which leads to differences in the grayscale of the image background and continuous changes in the grayscale of the weld zone, making defect recognition based on image threshold segmentation difficult. In terms of reducing the impact caused by background gray levels, X-ray radiography methodology [29] and infrared thermography [30] have been researched to increase the signal-to-noise ratio (SNR) of the weld region image, which can enhance the recognition rate of defects. In terms of image correction, a standard X-ray digital radiographic data acquisition procedure, along with optimized radiographic exposure conditions have been implemented to suppress structural noise [31].

In order to fulfill automatic defect detection in the lap joint of unequal-thickness plates, we first pre-processed the original radiograph to obtain an image in which the lap joint weld was vertically distributed. In subsequent work, a novel way of background removal is introduced to reduce the difficulty in defect recognition caused by variations in background grayscale. At the same time, several image processing methods such as noise suppression, image segmentation, and mathematical morphology have been introduced. Lastly, in the X-ray image of the lap joint of unequal-thickness plates, defects are automatically extracted.

## 2. Experimental Subjects and Equipment

The MU2000 X-ray real-time imaging system produced by the YXLON Company of Germany (Hamburg, Germany) was adopted. The weld specimens used in this work are two low-carbon steel plates with unequal thickness in the lap joint structure. The thicknesses of the plates are 4 mm and 2 mm, respectively. Figure 1a shows the geometric form of the specimen, and Figure 1b shows the appearance of the actual weldment.

Figure 2 shows the overall testing system combined with the defects testing methods used in this paper. In this paper, the weldment is placed on a stage with a three-axis rotation system and is radiographed by the X-ray source. The generated radiographic information is transmitted to the computer through the detector. Defects in the images received can be successively detected through a series of digital imaging processing techniques. These include image correction based on moment invariants, noise suppression based on wavelets, image segmentation based on thresholds, and mathematical morphology.

## 3. Methodology

### 3.1. Principle of X-Ray Radiographic Testing

In the radiographic method, radiation from a source is directed onto the specimen to be radiographed. The transmitted intensity through the specimen is recorded on a detecting medium, such as film, on the other side of the specimen. The intensity of the radiation beam decreases exponentially with the thickness *x* of the specimen. In radiographic testing, attenuation in the specimen can be calculated by using formula (1):(1)$$I=I_{0}B^{-\mu x}$$
where *I* is intensity of radiation emerging out of the specimen, *I*_0_ is intensity of radiation without a specimen, *μ* is linear attenuation coefficient per unit thickness and *B* is build-up factor which depends on the energy of the source, attenuating material, and its thickness.

### 3.2. X-Ray Radiographic Image Correction

To facilitate automatic defect detection, the distribution of weld seams in the radiograph is better to be vertical. However, during the process of specimen placing and clamping before X-ray radiography, the spatial position of the weld seam to be tested is uncertain due to factors such as the shapes of the specimen and fixture. Additionally, mechanical errors in the system’s transmission mechanism, among other factors, can easily cause specimens to be positioned variably, including weld seams offset from the center or vertically arranged welds having an inclination angle relative to gravity. As shown in Figure 3, there is a certain tilt angle α between the weld seam and gravity direction, affecting the effective implementation of subsequent image processing methods.

To eliminate this effect, a method based on moment invariants is used to calculate the inclination angle. Then, rigid transformation is introduced to translate the original radiograph, correct the inclination angle, and fulfill image correction.

#### 3.2.1. Moment Invariants

Moment invariants have been widely used in pattern recognition and digital image processing since M.K. Hu proposed the theorem in 1962. This research proved that moment invariants have the properties of rotation and translation invariance. To better study the various properties of moments, researchers have also given a variety of definitions of moments, such as Zernike moments, wavelet moments and so on.

For Hu moments, if the digital image function is piecewise continuous and has non-zero values in a finite part of the *x*, *y* plane, the existence of its $\;\left(p+q\right)th$ order moment $\;m_{pq}$ can be proven. It can also be proved that $m_{pq}$ is uniquely determined by f(x,y); conversely, f(x,y) can also be uniquely determined by $m_{pq}$. It is important to note that the assumption of finiteness is essential for the uniqueness theorem, otherwise the conclusion may not hold. The (*p* + *q*)*th* order moment in summation form can be defined as formula (2):(2)$$m_{pq}=\sum_{m=1}^{M}\sum_{n=1}^{N}x^{p}y^{q}f\left(x,y\right)$$
where (x,y) can be defined as pixel coordinate, f(x,y) is the grayscale function for (x,y). Formula (2) shows that it is easy to define the central moments in terms of moment invariants. It is well known that the central moment does not change under the translation of coordinates. Therefore, the $\left(p+q\right)th$ central moment of f(x,y) can be defined as formula (3):(3)$$\mu_{pq}=\sum_{m=1}^{M}\sum_{n=1}^{N}{\left(x-\bar{x}\right)}^{p}{\left(y-\bar{y}\right)}^{q}f(x,y)$$
where p,q=0,1,…, $\left(\bar{x},\bar{y}\right)$ are the barycentric coordinates, with $\bar{x}=m_{10}/m_{00}$ and $\bar{y}=m_{01}/m_{00}$. The first four orders of central moments can be deduced as follows: $\mu_{00}=m_{00}\equiv \mu$, $\mu_{01}=\mu_{10}=0$, $\mu_{20}=m_{20}-\mu \bar{x}$, $\mu_{11}=m_{11}-\mu \bar{y}$, $\mu_{02}=m_{02}-\mu \bar{y}$.

The $\left(p+q\right)th$ normalized central moment $\eta_{pq}$ can be defined as formula (4):(4)$$\eta_{pq}=\mu_{pq}/\mu_{00}^{r}$$
where r=1+(p+q)/2.

Any geometrical pattern or alphabetical character can always be represented by a density distribution function with respect to a pair of axes fixed in the visual field. In terms of X-ray radiographs, the grayscale level of the image can be regarded as the density distribution function of the pattern. Based on the uniqueness theorem mentioned above, the pattern can also be represented by its two-dimensional moments $m_{pq}$, with respect to a pair of fixed axes. Furthermore, if these central moments are normalized in size by using the similitude moment invariants, then the set of moment invariants can still be used to characterize the particular pattern. Normalized moments are independent of the pattern position in the visual field and are also independent of the pattern size. Therefore, the inclination angle can be calculated based on the invariant theorem.

For a two-dimensional image, *x* represents the gray level center of the image in the horizontal direction and y  represents the gray level center of the image in the vertical direction. The inclination angle θ that minimizes the second-order central moment can be expressed by formula (5):(5)$$\theta =\frac{1}{2}arctan\frac{2\mu_{11}}{\mu_{20}-\mu_{02}}$$

The *x*′ and *y*′ axes, determined by any values of θ satisfying formula (5), are called the principal axes of the pattern. With added restrictions, θ can be determined uniquely. Moments determined for such a pair of principal axes are independent of orientation. Subsequently, $\mu_{02}$ is the extension of the image in the horizontal direction; $\mu_{20}$ is the extension of the image in the vertical direction; and θ represents the inclination of the image. When $\mu_{11}>0$, the image is inclined to the upper left; otherwise, when $\mu_{11}<0$, the image is inclined to the upper right.

The result of the previous calculation, namely the value of the inclination angle θ, forms the basis for performing subsequent image processing methods. To fulfill image correction, a suitable translation algorithm is necessary.

#### 3.2.2. Rigid Body Translation

When correcting an image, it is often necessary to transform it accordingly. Rigid body translation means that the distance between two points in the transformed image remains the same as the previous image after transformation. Image contours provide important information for feature extraction. The contour of the specimen is regarded as a rigid body in order to utilize the properties, and the positioning information of the object can be represented by the calculation of the image contour features.

When the image to be processed is rotated and in need of correction, the rigid body transformation Formulas (6)–(8) can be used and are shown as follows:(6)$$\left[\begin{array}{c} x\\ y \end{array}\right]=\left[\begin{array}{cc} cos∆\theta & -sin∆\theta \\ sin∆\theta & cos∆\theta \end{array}\right]\left[\begin{array}{c} x\\ y \end{array}\right]+\left[\begin{array}{c} ∆x\\ ∆y \end{array}\right]$$
(7)$$∆\theta =\theta -∆\theta_{m}$$
(8)$$\left[\begin{array}{c} ∆x\\ ∆y \end{array}\right]=\left[\begin{array}{c} x_{c}^{\prime }\\ y_{c}^{\prime } \end{array}\right]-\left[\begin{array}{c} x_{c}\\ y_{c} \end{array}\right]$$
$\left(x_{c},y_{c}\right)$ is the centroid coordinate of the original image, $\theta_{m}$ is the angle between the main axis and the positive direction of the *x*-axis; $\left(x_{c}^{\prime },y_{c}^{\prime }\right)$ is the centroid coordinate of the translated image; and ∆x is the displacement required to register the previous image with the template image, where θ is the angle between the main axis and the positive direction of the *x*-axis. Figure 4 shows the schematic diagram of the overall image correction steps.

### 3.3. Defect Detection in X-Ray Radiographics

The transitional region refers to the area between the target and the background. It is a special region, that not only can separate different regions based on boundary attributes, but its width and area are also both non-zero values. Due to the change of thickness, the radiograph of lap joints presents continuous grayscale distribution in the welding area, which is in line with the characteristics of the transitional region. Compared with the non-transitional region, the changes in the grayscale of the transitional region are more frequent and dramatic. These results are caused by variations of thickness in the fillet weld seam, defects and noise. Given a radiograph with limited grayscale level, the targets segmented can be divided into extended targets and weak targets, according to their size proportions. In this case, the weld seam is the extended target, and the defect is the weak target. The sudden change of grayscale level in the continuous transitional region represents the existence of defects.

After transforming the original radiograph, the weld seam is vertically distributed along the longitude of the image. Additionally, the horizontal grayscale distribution changes regularly according to the change in thickness. In order to identify defects, noise suppression based on wavelet threshold is carried out first as image pre-processing. Based on this, the simulation of the background image can be obtained. The purpose of simulating the background is to eliminate the sudden change in linear grayscale level caused by the existence of defects. Afterward, the method of least squares is used to fit the grayscale curve of the weld seam in the processed image. Then, the foreground image is obtained by subtracting the transformed image from the fitted background simulation. Finally, the foreground image is processed based on iterative segmentation and mathematical morphology, and the weak targets, namely the defects in the X-ray radiograph, can be finally extracted. Figure 5 shows the flow diagram of digital image processing.

#### 3.3.1. Noise Suppression

The industrial television inspection system, based on the image intensifier, can display the defect status of the specimen in real time, making it suitable for dynamic inspection. However, as the X-ray radiation goes through the image intensifier, the information of the radiograph is easily disturbed. Therefore, the test results of the real-time detection system present low sensitivity and high noise levels. In terms of noise suppression, methods based on wavelets and wavelet transformation have been widely used. At present, the commonly used noise suppression methods include wavelet modulus maxima, nonlinear wavelet transform thresholding, and noise reduction based on wavelet transformation. Threshold filtering is the simplest filtering method. Its basic principle is to compare the amplitude of the wavelet coefficient with a preset threshold. If the wavelet coefficient is lower than the preset threshold, it is set to 0; however, if the wavelet coefficient is higher than the preset threshold, it is retained. The wavelet threshold method is a simple method with low computational cost which has great potential practicability in image processing. The implementation procedure of the algorithm is as follows:Select the appropriate wavelet and wavelet decomposition level N, each will obtain the wavelet coefficients of low-frequency components and high-frequency components, that is, approximation signals and detail signals. Continue to perform wavelet decomposition on approximation signals and obtain a set of wavelet coefficients;Perform quantization processing on the wavelet coefficients obtained through decomposition, based on the threshold method, to estimate wavelet coefficients;Use the estimated wavelet coefficients to perform inverse wavelet transformation, which also known as wavelet reconstruction, to obtain a noise-suppressed image.

#### 3.3.2. Background Removal

Several regions account for the change of thickness, including the thick plate region, the overlapping region of the two plates, the non-linear thickness change in the weld seam region, and the thin plate region. A novel method of background removal is therefore introduced. The background image is first calculated based on statistics. Afterwards, the foreground image which contains defect information, is obtained by subtracting the original image with the stimulated background image. Figure 6a shows the target region of the joint. Figure 6b shows the continuous change in the grayscale of the background of the image. As the grayscale of the weld area changes continuously in the x direction, the grayscale of possible defects may be submerged in the grayscale value of the background image, as shown in Figure 6c,d. The unique grayscale distribution of the image brings difficulties in detecting defects. Therefore, the method of background image filtering is suitable for automatic defect detection. Assuming that the original grayscale distribution of the radiograph is $\;P_{\left(i,j\right)}$, the line grayscale distribution $b_{i}$ of the background image $B_{\left(i,j\right)}$ is obtained by using the matrix column vector mean, as shown in formula (9).
(9)$$b_{i}=\frac{1}{y}\sum_{j=1}^{y}P_{\left(i,j\right)},i=1,2,3,\ldots ,x$$

The formula for calculating the grayscale $F_{\left(i,j\right)}$ of foreground image is shown in formula (10).
(10)$$F_{\left(i,j\right)}=P_{\left(i,j\right)}-B_{\left(i,j\right)},i=1,2,3,\ldots ,x,j=1,2,3,\ldots ,y$$

Figure 6f shows that after background removal, the grayscale distribution of the foreground image contains defects, noise, and a small proportion of background resistance.

#### 3.3.3. Image Segmentation

According to the grayscale histogram of the image, the optimal threshold can be calculated using mathematical statistics. Also, the iterative method is used to solve the image segmentation threshold and is as follows:Set the program terminal parameter $\lambda_{0}$. At the same time, select a suitable threshold value $\lambda_{1}$ based on grayscale distribution;Segment the image with $\lambda_{1}$ and all pixels can be divided into two sets: set A includes pixels with greater grayscale value than $\lambda_{1}$ and set B includes pixels with smaller grayscale values than $\lambda_{1}$;Calculate the average grayscale value of each set and get $\tau_{1}$ and $\tau_{2}$, the new threshold $\lambda_{2}$ can be obtained using formula (11):(11)$$\lambda_{2}=\left(\tau_{1}+\tau_{2}\right)/2$$Terminate the program when the optimal threshold is found, which satisfies the constrain in formula (12):(12)$$\left|\lambda_{1}-\lambda_{2}\right|<\lambda_{0}$$

Otherwise, repeat procedure (1)~(3) until the optimal threshold is found.

#### 3.3.4. Mathematical Morphology

Post-processing based on mathematical morphology is performed after image segmentation, using structural units to assign the values to the central element. The basic operations are erosion and dilation. The opening operation and closing operation are two important operations in morphology. The opening operation is to erode and then dilate, which can generally smooth the contour of the image. Contrary to the opening operation, the closing operation is dilation first and then erosion, which can fuse the narrow gaps in the image and fill the gaps in the contour. For example, if *A* is a binarized ray image, and *B* is a selected structuring element, then *A* is opened by structural element *B* which is defined as formula (13):(13)A∘B=(AΘB)⊕B
where (AΘB) is defined as erosion in formula (14);
(14)$$(A\Theta B)=\left\{\left.x\right|{\left(B\right)}_{x}\subseteq A\right\}$$

Similarly, the closing operation is defined as formula (15);
(15)A•B=(A⊕B)ΘB
where (A⊕B) is defined as dilation in formula (16);
(16)$$A⊕B=\left\{\left.x\right|\left[{\left(\hat{B}\right)}_{X}⋂A\right]\subseteq \Phi \right\}$$

In this paper, the structural unit *B* is defined as formula (17);
(17)$$B=\left|\begin{array}{ccc} 0 & 1 & 0\\ 1 & 1 & 1\\ 0 & 1 & 0 \end{array}\right|$$

## 4. Experimental Results and Analysis

### 4.1. Image Pre-Processing

The original radiograph obtained is shown in Figure 7a. The weld occupies a large proportion of the whole image, and the welding seam area is inclined at a small angle. The thickness of the weld varies nonlinearly and continuously along the thickness direction of the plates, resulting in a continuous change in the grayscale of the acquired radiograph. As welding speed varies, alterations in wire feeding speed and other process factors lead to changes in the amount of cladding, affecting the width and height of the weld. Furthermore, the presence of defects significantly alters the grayscale of the radiograph, making the overall scenario more complex. This brings difficulties to automatic defect identification. The weld area is initially obtained through ROI recognition on the original radiograph for subsequent processing.

Firstly, the radiograph is binarized. In this paper, image binarization employs an iterative thresholding approach to determine a threshold value, which is then utilized to isolate the weld area from the analyzed image. The extraction of weld edges is facilitated by the Canny algorithm, which leverages the Laplacian operator to discern between prominent and subtle edges by applying two distinct thresholds. This methodology selectively includes weak edges only when they are contiguous to stronger ones, thereby minimizing noise interference and enhancing the detection of genuine, faint edges compared to alternative methods. In the binarized image extracted from the weld area, the weld inclination angle is calculated to be 3.1° according to the invariant moment theory, and the result is presented in Figure 7b. On this basis, the image is corrected, and the results are shown in Figure 7c,d. It can be seen from the correction results that the corrected image maintains the characteristics of the original radiograph, and detailed information is not lost during the rotation process. This provides the basis for the later image processing, defect recognition, and extraction.

### 4.2. Automatic Defects Detection

The corrected image shown in Figure 7d is subjected to wavelet-based noise suppression processing to obtain Figure 8a, which removes high-frequency noise components in the image. The noise-suppressed image is used to estimate the image background and obtain the background image shown in Figure 8b.

The foreground image was obtained by subtracting the corrected image and the background image shown in Figure 7d and is shown in Figure 8c. The foreground image is composed of defect images, noise points, and residual background components. The foreground image is further processed by image threshold segmentation based on the iterative method, and the background residue and noise points in the image are filtered out to obtain Figure 8d. Finally, the operations of expansion, corrosion, and opening and closing based on mathematical morphology are performed to filter out isolated noise points, separate adhesion targets, and realize automatic defect identification. The results are shown in Figure 8e.

The method proposed enables automatic defect recognition, sizing, measurement, and localization, utilizing image processing techniques without the need for extensive samples or training, as required by machine learning.

## 5. Discussion

A novel background removal method based on image processing is proposed in this paper. A radiographic image can be regarded as a matrix, where the values of elements in row vectors and column vectors represent the grayscale at different positions. The method proposed in this paper is as follows: first, the row vectors in the welding joint section of the radiographic film are statistically analyzed, and the average row vector is obtained as the simulated background image; then, the original image is subtracted from the simulated background image to obtain the foreground image we are interested in, the defect image.

The reasons for doing so are twofold. Firstly, the defects in the weld are mainly a few small, dispersed porosity defects. Therefore, when calculating the average grayscale of horizontal lines, the grayscale variation caused by the defects has little impact. Secondly, since the specimen is automatically welded, the reinforcement height along the longitudinal direction of the lap weld varies slightly, which means that the attenuation of the X-ray penetration varies slightly. Therefore, the trends of grayscale change in row vectors are similar, allowing for the use of averaging for statistical analysis.

In conclusion, when the inspected object has drastic and irregular thickness changes or a large number of defects with large sizes, we believe that the background removal method proposed in this paper may result in a decrease in detection accuracy. However, unlike defect detection methods based on neural networks which demand a large amount of sampling and training, the method proposed is efficient and easy to modify according to different structures.

## 6. Conclusions

The characteristics of digital X-ray images of lap weld structures with unequal thickness plates are analyzed and researched. Firstly, the variation in the thickness of the workpiece leads to differences in the grayscale of the image background and continuous changes in the grayscale of the weld zone. Secondly, the position of the weld seam in the radiograph is not vertical and the place of it is uncertain.To facilitate automatic defect detection, the distribution of weld seam in the radiograph is first preprocessed to be vertical. First, the moment invariants method is introduced to calculate the inclination angle. Then, rigid body transformation is applied to fulfill image correction. The preprocessing of the original radiograph provided a solid foundation for subsequent work.Based on preprocessing, a background removal method through background simulation was applied to the image. This resulted in obtaining the radiographic foreground image through background removal.Through threshold segmentation and mathematical morphology, a binary image of defects is obtained. The automatic recognition of defects in the X-ray radiograph of the lap joint with an unequal thickness plate was achieved.The proposed method enables automatic recognition, sizing, measuring, and locating of defects. It is an image processing-based method that does not require a large number of samples and training, as machine learning methods do.

## Figures and Tables

**Figure 1 materials-17-05463-f001:**
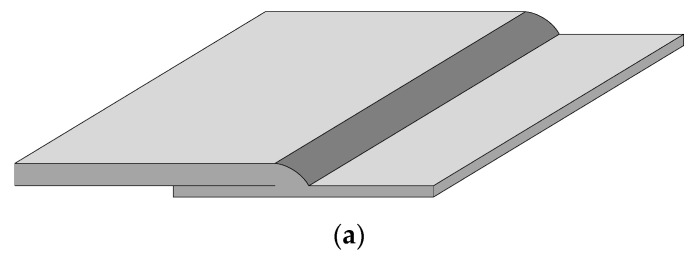
Preparation for weld specimen: (**a**) Geometric form of the joint, (**b**) Appearance of the weld specimen.

**Figure 2 materials-17-05463-f002:**
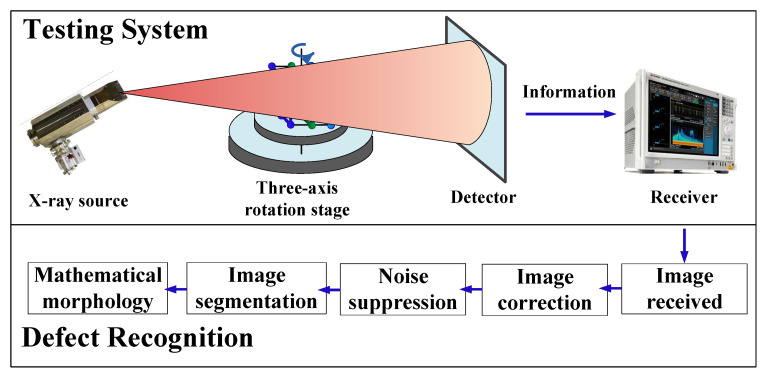
Overall testing system and defect testing methods.

**Figure 3 materials-17-05463-f003:**
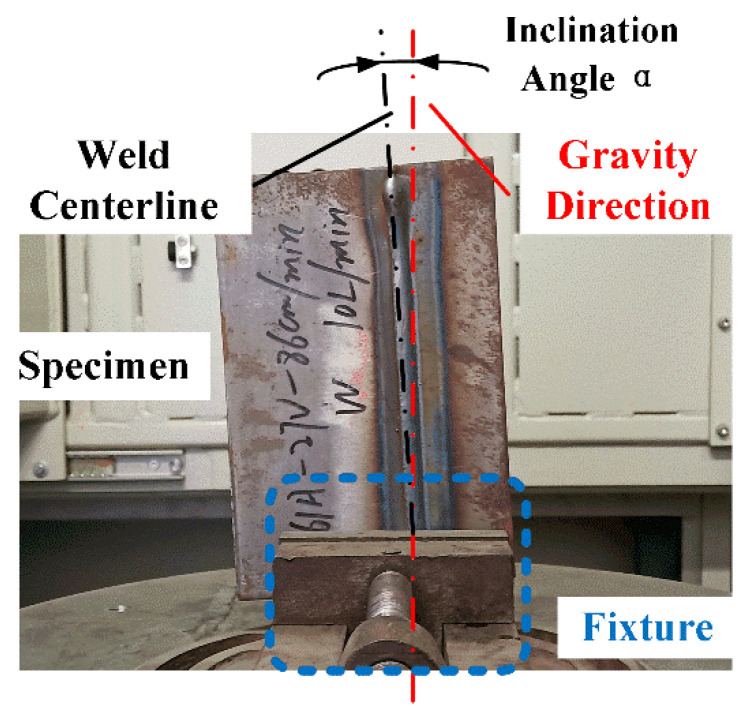
Positioning of the weld under testing.

**Figure 4 materials-17-05463-f004:**
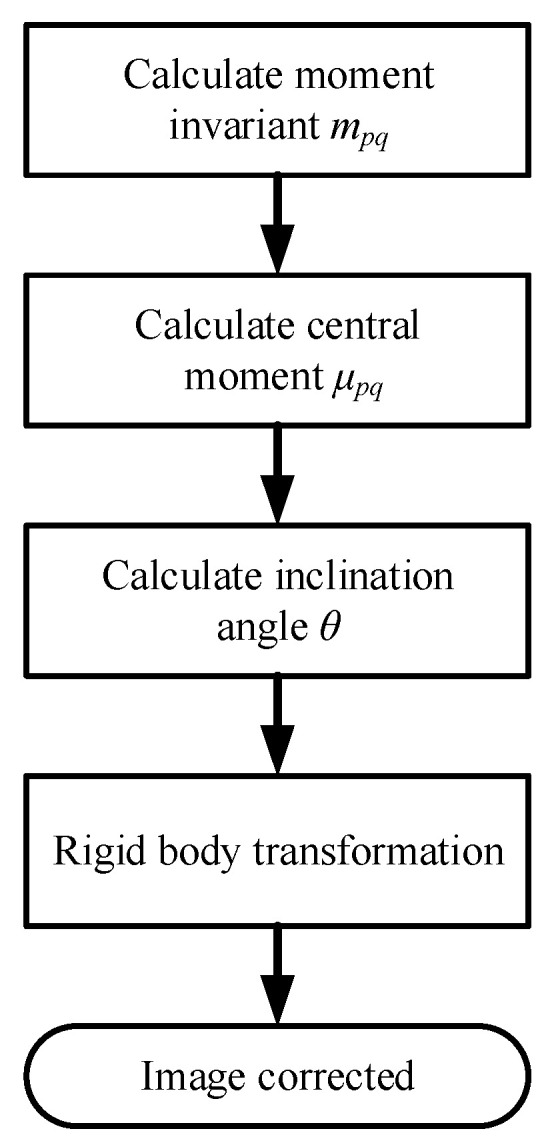
Image correction steps.

**Figure 5 materials-17-05463-f005:**
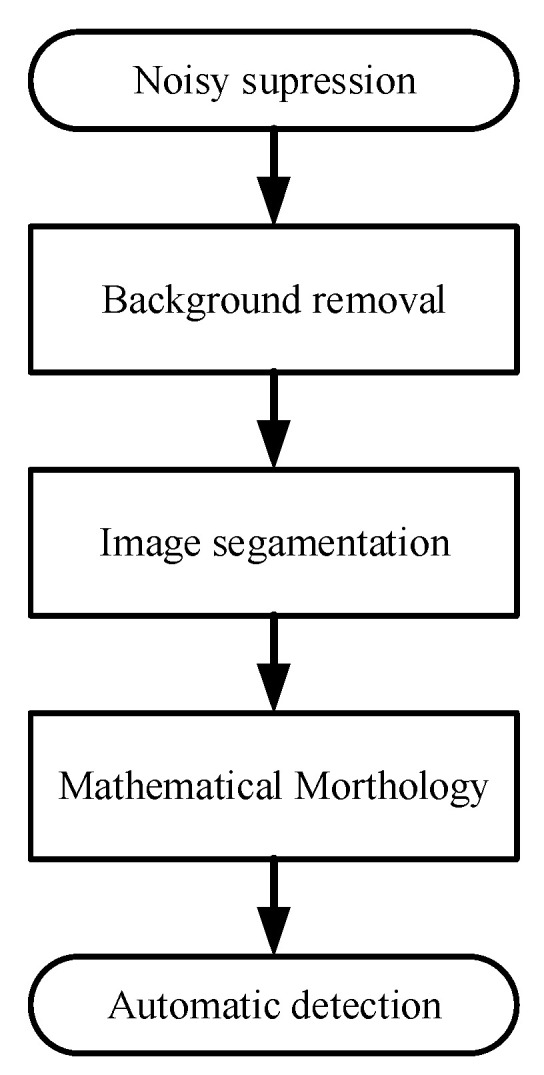
Digital image processing for defect detection.

**Figure 6 materials-17-05463-f006:**
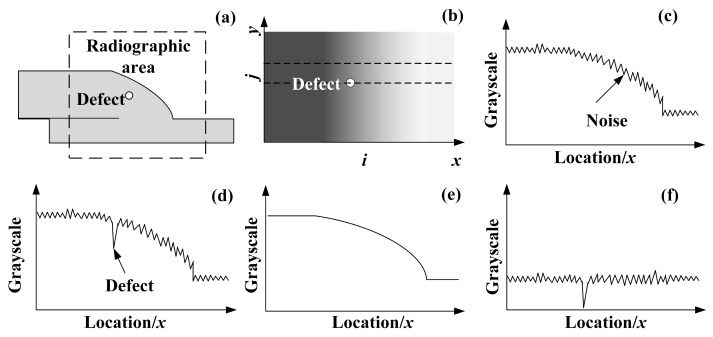
Background removal. (**a**) Cross-section of the lap joint. (**b**) Grayscale distribution of the radiograph. (**c**) Linear grayscale distribution without defect. (**d**) Linear grayscale distribution with defect. (**e**) Linear grayscale distribution of background. (**f**) Linear grayscale distribution of foreground.

**Figure 7 materials-17-05463-f007:**
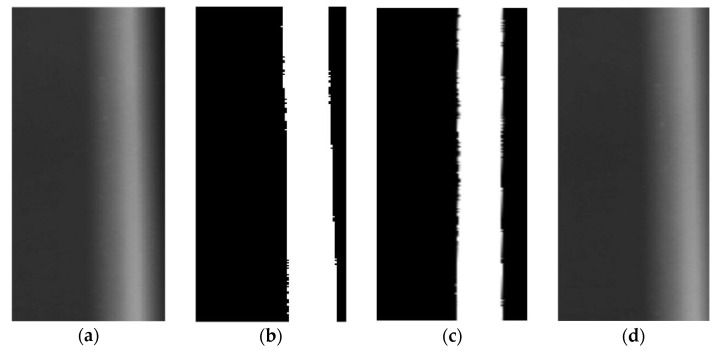
Image corrections: (**a**) Original radiograph, (**b**) Contour extraction, (**c**) Image correction, (**d**) Image corrected.

**Figure 8 materials-17-05463-f008:**
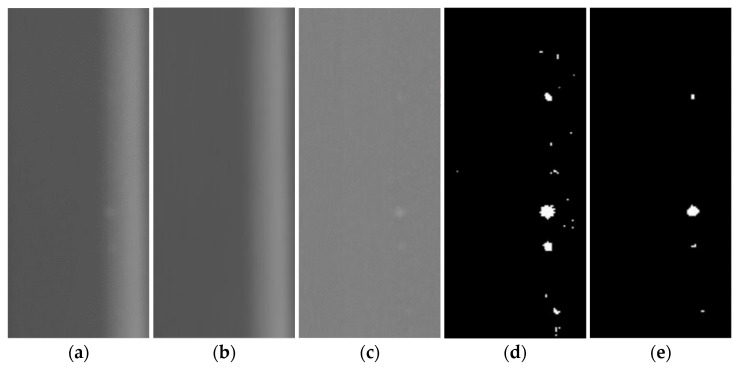
Defect detection images: (**a**) Noise suppression, (**b**) Background image, (**c**) Foreground image, (**d**) Image segmentation, (**e**) Mathematical morphology.

## Data Availability

The original contributions presented in the study are included in the article, further inquiries can be directed to the corresponding author.

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
