# Peer review of "Automatic Defects Recognition of Lap Joint of Unequal Thickness Based on X-Ray Image Processing"

_materials, 2024, doi:10.3390/ma17225463_

Round 1
Reviewer 1 Report
Comments and Suggestions for Authors
In the peer-reviewed manuscript: Automatic Defects Recognition of Lap Joint of Unequal Thick- ness Based on X-ray Image Processing.
an X-ray image correction method based on invariant moments is presented to solve this problem. An automatic defect detection method combining image noise suppression, substrate removal, image segmentation and mathematical morphology is introduced. The title is aptly presented, as is the abstract. Introdaction is quite laconic, and the citations given, refer to obvious information, e.g., X-ray inspection has become commonplace in welding, additive manufacturing and many other manufacturing processes [2-4]. Similarly: X-ray inspection is the most intuitive approach to non-destructive testing (NDT) of flat welded components [13-16], pipelines [17,18] to avoid pipeline safety accidents. In my opinion, this section lacks details, disquisitions that distinguish a scientific article from a school textbook.
Despite the above, the authors have demonstrated the need for the research, which has been fully justified. What is definitely missing is a clear research plan, where step by step the authors present what they will research, with what methods, with what aprature and what samples. The samples should be described in detail. In the following section, the authors presented some relationships - expressions that increase the value of the work. Image correction steps (fig. 4) is simple and intuitive. The idea of background removal is very interestingly presented in Fig. 6. background.
The authors, did not have a discussion, which is important. I think the authors should show which centers, which researchers are conducting similar studies and compare their results with theirs. The form of the manuscript presented, pins a research report or a book excerpt and not a scientific publication. It is worth bearing in mind that although the authors have touched on an important topic, a topical subject, more people in the world are working on these issues and it is worth referring to their results.
Author Response
Dear Reviewer,
I am Wang Ziming, the author of the paper. First of all, I would like to express my sincere gratitude for your valuable suggestions on the revision of our article. My co-authors and I have found your comments to be extremely insightful and have made modifications accordingly to each of your proposals. Below are the specific changes made to the article.
Comment1:“Introdaction is quite laconic, and the citations given, refer to obvious information, e.g., X-ray inspection has become commonplace in welding, additive manufacturing and many other manufacturing processes [2-4]. Similarly: X-ray inspection is the most intuitive approach to non-destructive testing (NDT) of flat welded components [13-16], pipelines [17,18] to avoid pipeline safety accidents. In my opinion, this section lacks details, disquisitions that distinguish a scientific article from a school textbook.”
Response1:
- According to your advice, in Introduction line26-27, we add several explanations about the use of X-ray examination in different types of welding, including the arc welding and fusion welding and so on. Also, we further explain the use of X-ray examination in these manufacturing processes mainly focus on defect detection.
- In Introduction line 34-38, we added some details about the different uses of X-ray inspection, such as non-destructive testing, material evaluation and residual stresses analysis. We further explained that the purpose of conducting these X-ray based techniques is to avoid safety accidents and evaluate the quality of welds.
Comment2: “What is definitely missing is a clear research plan, where step by step the authors present what they will research, with what methods, with what aprature and what samples.”
Response2:
We further introduced the set of experiment in chapter2 “Experimental subjects and equipment”, in which we mentioned the X-rays test system, weld specimen and overall research plan step by step in Fig2.
Additionally, we have revised and added content to parts of the article.
- In the abstract, we have emphasized the background removal method proposed in the article and evaluated its effectiveness in detecting pore-like defects in automatic welding seams.
- To make the expression more scientific and specific, we have supplemented the research content of the references cited in introduction.
- In order to enable researchers from different fields to better understand the content, we have also supplemented the research status of methods related to this article.
- At the end of the introduction, we underlined the background removal method proposed in this paper.
- In 3.2, which deals with background removal, we have supplemented the overall idea of the method.
- Additionally, we have added Chapter 5 to discuss the applicability, advantages, and disadvantages of the proposed method.
Comment3: “The authors, did not have a discussion, which is important. I think the authors should show which centers, which researchers are conducting similar studies and compare their results with theirs.”
Response3:
We have added Chapter 5 to the article as the content for discussion. In this section, we mainly discuss the application conditions and scope of the proposed background removal method.
- We believe that the method can effectively detect dispersed pore-like defects in automatic welding seams, and the automatic defect recognition does not require a large number of samples for learning and training. It can be flexibly modified according to the shape and size of the actual workpiece.
- However, when the thickness of the inspected specimen varies drastically or when there are a large number of defects or relatively large defects, the method proposed in the article may result in a decrease in detection accuracy.
Thank you once again for your valuable suggestions on this paper.
Best regards,
Wang Ziming
Reviewer 2 Report
Comments and Suggestions for Authors
This paper requires major revisions before it can be considered for publication. While the topic of automatic defect detection in X-ray radiographs of lap joints is relevant and promising, there are several issues that need to be addressed. These include the need for clearer explanations of the methods used, stronger validation of the results, and comparisons with alternative approaches. Additionally, the structure and presentation of the methodology require more detail and clarity to ensure the work is accessible to readers from diverse academic backgrounds.
Here are some academic-level questions to guide revisions and further development of the paper:
-
The paper discusses the use of invariant moments to correct X-ray images for automatic defect detection. Could you explain the principle behind invariant moments, and why this method is particularly suitable for correcting the orientation of the weld seam in X-ray radiographs of lap joints?
-
The authors highlight the challenge of varying grayscale levels due to thickness variations in the lap joint plates. How does the proposed background removal method handle this issue, and what are the limitations of this approach when applied to more complex structures or materials?
-
The paper suggests that the proposed image processing-based defect detection method does not require a large dataset for training, unlike machine learning models such as CNNs. Can you compare the potential advantages and disadvantages of this method in terms of accuracy, scalability, and application across different industrial use cases?
-
Defect recognition in plates of unequal thickness presents unique challenges due to the non-uniform distribution of gray levels and the inclination of weld seams. What are the key limitations in the proposed method for handling these challenges, and how might future developments in image processing or machine learning further improve defect detection in such cases?
- Furthermore, more references need to be included in the introduction section, especially when discussing the use of Convolutional Neural Networks (CNNs) and automated decision-making processes. Relevant papers that can be added include "Cognitive State Classification Using Convolutional Neural Networks on Gamma-Band EEG Signals", published in Applied Sciences, 2024, and "Improved Diagnostic Process of Multiple Sclerosis Using Automated Detection and Selection Process in Magnetic Resonance Imaging", published in Applied Sciences, 2017. These references will strengthen the discussion around automation and modern image processing techniques in defect detection.
Author Response
Dear Reviewer,
I am Wang Ziming, the author of the paper. First of all, I would like to express my sincere gratitude for your valuable suggestions on the revision of our article. My co-authors and I have found your comments to be extremely insightful. Here are the replies to your questions and several modifications have been made accordingly to each of your proposals.
Comment1: The paper discusses the use of invariant moments to correct X-ray images for automatic defect detection. Could you explain the principle behind invariant moments, and why this method is particularly suitable for correcting the orientation of the weld seam in X-ray radiographs of lap joints?
Response1:
- The moment invariant originated from probability density function in probability theory and mathematical statistics. In line 121-165 we explained the principle behind invariant moments.
- As for X-ray images, the distribution of greyscale can also be interpreted as a probability density function.
- After several derivations, the Hu moments have been proved to have geomatic invariance such as translation and rotation. So the properties of moment invariant indicates that it is particularly suitable for correcting the inclination angle of weld in radiograph, as in 2 X-ray radiographic image correction, line 100-108.
Comment2: The authors highlight the challenge of varying grayscale levels due to thickness variations in the lap joint plates. How does the proposed background removal method handle this issue, and what are the limitations of this approach when applied to more complex structures or materials?
Response2:
- We further explained our approach and added the overall idea to deal with the varying background grayscale in line 233-238.
- A radiographic image can be regarded as a matrix, where the values of elements in row vectors and column vectors represent the grayscale at different positions. The method proposed in this paper is as follows: first, the row vectors in the welding joint section of the radiographic film is statistically analyzed, and the average row vector is obtained as the simulated background image; then, the original image is subtracted from the simulated background image to obtain the foreground image we are interested in, the defect image.
- The reasons for doing so are twofold: 1) We found that the defects in the weld of unequal-thickness steel plates are relatively few and their longitudinal distribution size is small. Therefore, when calculating the average grayscale of horizontal lines, the grayscale variation caused by defects has little impact. 2) We found that the reinforcement height along the longitudinal direction of the lap weld varies slightly, which means that the attenuation when the X-rays penetrate the weld varies slightly. Therefore, the trends of grayscale change in row vectors are similar, allowing for the use of averaging for statistical analysis. When the inspected object has drastic and irregular thickness changes or a large number of defects with large size, we believe that the background removal method proposed in this paper may result in a decrease in detection accuracy.
Comment3: The paper suggests that the proposed image processing-based defect detection method does not require a large dataset for training, unlike machine learning models such as CNNs. Can you compare the potential advantages and disadvantages of this method in terms of accuracy, scalability, and application across different industrial use cases?
Response3:
To discuss the advantages and disadvantages of the proposed method to CNN based method, we add section 5 as Discussion in line 329-350.
In terms of accuracy, applicability and potential, we believe that the automatic defect recognition method based on deep learning has more advantages than that based on image processing.
- The advantage of the method proposed in this paper is its high efficiency, as it can achieve automatic defect recognition through image processing algorithms without the need to collect a large amount of data.
- Meanwhile, in applications, the method proposed can be flexibly modified according to different structures.
Comment4:Defect recognition in plates of unequal thickness presents unique challenges due to the non-uniform distribution of gray levels and the inclination of weld seams. What are the key limitations in the proposed method for handling these challenges, and how might future developments in image processing or machine learning further improve defect detection in such cases?
Response4:
- In Introduction line 59-69, we added several citations to introduce the key limitation of current researches.
- When dealing with a large number of samples, automatic defect recognition methods based on deep learning are most suitable. In recent years, the application potential of artificial intelligence models has been well-validated in fields such as medicine.
- Through the references you provided, we have gained a deeper understanding of these research findings. However, in practical engineering, to meet various needs, some structures can be relatively novel. Automatic defect recognition for these new structures mainly faces the challenge of limited sample sizes.
- Therefore, this paper adopts an image processing-based method to achieve automatic defect recognition for lap welds of unequal thickness plates. Additionally, the method developed in this paper can be quickly adjusted according to the actual size and form of the workpiece, with lower requirements for time and sample size compared to deep learning methods.
Comment5: Furthermore, more references need to be included in the introduction section, especially when discussing the use of Convolutional Neural Networks (CNNs) and automated decision-making processes. Relevant papers that can be added include "Cognitive State Classification Using Convolutional Neural Networks on Gamma-Band EEG Signals", published in Applied Sciences, 2024, and "Improved Diagnostic Process of Multiple Sclerosis Using Automated Detection and Selection Process in Magnetic Resonance Imaging", published in Applied Sciences, 2017. These references will strengthen the discussion around automation and modern image processing techniques in defect detection.
Response5: We have carefully read the articles you recommended and believe that the content can help us enrich our paper, making it easier for scholars from different fields to understand. After discussion, we have decided to cite "Cognitive State Classification Using Convolutional Neural Networks on Gamma-Band EEG Signals", published in Applied Sciences, 2024 and “Human gamma-frequency oscillations associated with attention and memory”, published in “Trends Neurosci”,2007 to enrich the introduction, adding them as supplements to the background of deep learning.
In addition, we have revised and added content to parts of the article.
- In the abstract, we have emphasized the background removal method proposed in the article and evaluated its effectiveness in detecting pore-like defects in automatic welding seams.
- To make the expression more scientific and specific, we have supplemented the research content of the references cited in introduction.
- In order to enable researchers from different fields to better understand the content, we have also supplemented the research status of methods related to this article.
- At the end of the introduction, we underlined the background removal method proposed in this paper. In 3.2, which deals with background removal, we have supplemented the overall idea of the method.
- Additionally, we have added Chapter 5 to discuss the applicability, advantages, and disadvantages of the proposed method.
Thank you once again for your valuable suggestions on this paper.
Best regards,
Wang Ziming
Reviewer 3 Report
Comments and Suggestions for Authors
Dear authors,
I have been invited to review the paper “Automatic Defects Recognition of Lap Joint of Unequal Thickness Based on X-ray Image Processing”.
Further to a careful review of your paper, I would like to make the following comments for your consideration:
• Abstract
You might wish to consider revise the text and ensure it becomes self-standing. This would help increase the attractiveness to potential readers. Please underline the novelty of your work and provide the context of your work.
• Introduction
While you refer vaguely to some previous studies, e.g., lines 44-46, I would like to invite you to conduct and present a fully comprehensive literature review of the research work done previously, identify existing research/knowledge gaps which should in turn lead you to explain the rationale for your research work.
You might also wish to introduce the research approach you will us in this piece of work.
To facilitate the understanding of the audience of your paper, please develop and insert a comprehensive theoretical framework for your research. You might wish to use a graphical representation to facilitate the understanding of your work and research objectives.
2. X-ray inspection system and weld specimen preparation
Please ensure a smooth transition from one section to another. Is this section part of the methodology? Currently, it is not clear to me how this fits into a research paper, immediately after the introduction and with this title.
3. Methodology
Please indicate whether the methodology you are using is part of your contribution and original. Otherwise, please indicate who has developed it and provide relevant references.
5. Conclusion
Please indicate the limitations of your research and based on them put forward ideas for future research.
Please make sure you underline the novelty and contributions of your work in the conclusions.
On a general note, I would like to recommend that you substantially review the manuscript. Currently, it presents a case study/experiment. While the latter might be interesting, it is important to ensure that your manuscript is written in the form of a research paper. The above recommendations might help you revise the document.
Best regards,
Anonymous reviewer
Author Response
Dear Reviewer,
I am Wang Ziming, the author of the paper. First of all, I would like to express my sincere gratitude for your valuable suggestions on the revision of our article. My co-authors and I have found your comments to be extremely insightful. Here are the replies to your questions and several modifications have been made accordingly to each of your proposals.
Comment1: You might wish to consider revise the text and ensure it becomes self-standing. This would help increase the attractiveness to potential readers. Please underline the novelty of your work and provide the context of your work.
Response1:
- In Abstract, according to your advise we underlined the background removal method proposed in this paper and introduced our evaluation on this method. The detail is as following:
- “In addition, a novel way of background removal method based on image processing is introduced to reduce the difficulty of defect recognition caused by variations in grayscale. At the same time, an automatic defect detection method combined of image noise suppression, image segmentation and mathematical morphology is adopted. The results show that the proposed method can effectively recognize the gas pores in automatic welded lap joint of unequal thickness and it is suitable for automatic detection.”
Comment2: To facilitate the understanding of the audience of your paper, please develop and insert a comprehensive theoretical framework for your research.
Response2:
- In Introduction, we supplemented the work content of partially ambiguous references. Also to better present our work to scholar from different fields, we added several citations.
- We further introduced the set of experiment in chapter2 “Experimental subjects and equipment”, in which we mentioned the X-rays test system, weld specimen and comprehensive research plan step by step in Fig2.
Comment3: Please ensure a smooth transition from one section to another. Is this section part of the methodology? Currently, it is not clear to me how this fits into a research paper, immediately after the introduction and with this title.
Response3:
- In addition, we added a transition paragraph (line 70-77) at the end of Introduction to transit to section 2.
- In section2, we changed the title into “Experimental subjects and equipment” according to your advice. We found the previous title lack a smooth transition from the Introduction.
Comment4: Please indicate whether the methodology you are using is part of your contribution and original. Otherwise, please indicate who has developed it and provide relevant references.
Response4:
The methods involved in this paper are moment invariant, noisy suppression, background removal, image segmentation and mathematical morphology.
- As for moment invariant, we clarified that it originated from Hu in 1964 in line 97-100.
- As for noisy suppression, image segmentation and mathematical morphology, these methods have been widely used in digital image processing.
- As for background removal, we underline that the method is originally proposed by the authors in line 235-237, we have also supplemented the overall idea of the method.
Comment5: Please indicate the limitations of your research and based on them put forward ideas for future research.
Response5:
We have added Chapter 5 in line 330-350 to discuss the applicability, advantages, and disadvantages of the proposed method.
For future research, we provide several ideas as following:
- when the inspected object has drastic and irregular thickness changes or a large num-ber of defects with large sizes, we believe that the background removal method pro-posed in this paper may result in a decrease in detection accuracy.
- unlike defect detection methods based on neural network which demands a large amount of sampling and training, the method proposed is efficient and easy to be modified ac-cording to different structure.
Additionally, we also added revisions as following:
- We have revised and added content to parts of the article. In the abstract, we have emphasized the background removal method proposed in the article and evaluated its effectiveness in detecting pore-like defects in automatic welding seams.
- To make the expression more scientific and specific, we have supplemented the research content of the references cited in introduction.
- In order to enable researchers from different fields to better understand the content, we have also supplemented the research status of methods related to this article.
- At the end of the introduction, we underlined the background removal method proposed in this paper. In 3.2, which deals with background removal, we have supplemented the overall idea of the method.
- Additionally, we have added Chapter 5 to discuss the applicability, advantages, and disadvantages of the proposed method.
Thank you once again for your valuable suggestions on this paper.
Best regards,
Wang Ziming
Round 2
Reviewer 2 Report
Comments and Suggestions for Authors
The new version can be published.
Reviewer 3 Report
Comments and Suggestions for Authors
Dear authors,
Thank you for the revised version of your manuscript.
Best regards,
Anonymous reviewer